# Structure-Based High-Throughput Virtual Screening and Molecular Dynamics Simulation for the Discovery of Novel SARS-CoV-2 NSP3 Mac1 Domain Inhibitors

**DOI:** 10.3390/v15122291

**Published:** 2023-11-22

**Authors:** Behnaz Yazdani, Hajar Sirous, Simone Brogi, Vincenzo Calderone

**Affiliations:** 1Bioscience Department, Faculty of Science and Technology (FCT), Universitat de Vic—Universitat Central de Catalunya (Uvic-UCC), 08500 Vic, Spain; behnazyazdani6@gmail.com; 2Bioinformatics Research Center, School of Pharmacy and Pharmaceutical Sciences, Isfahan University of Medical Sciences, Isfahan 81746-73461, Iran; 3Department of Pharmacy, University of Pisa, Via Bonanno 6, 56126 Pisa, Italy; vincenzo.calderone@unipi.it

**Keywords:** nonstructural proteins, NSP3, macrodomain, Mac1, ADRP, ADP ribose phosphatase, SARS-CoV-2, drug discovery

## Abstract

Since the emergence of SARS-CoV-2, many genetic variations within its genome have been identified, but only a few mutations have been found in nonstructural proteins (NSPs). Among this class of viral proteins, NSP3 is a multidomain protein with 16 different domains, and its largest domain is known as the macrodomain or Mac1 domain. In this study, we present a virtual screening campaign in which we computationally evaluated the NCI anticancer library against the NSP3 Mac1 domain, using Molegro Virtual Docker. The top hits with the best MolDock and Re-Rank scores were selected. The physicochemical analysis and drug-like potential of the top hits were analyzed using the SwissADME data server. The binding stability and affinity of the top NSC compounds against the NSP3 Mac1 domain were analyzed using molecular dynamics (MD) simulation, using Desmond software, and their interaction energies were analyzed using the MM/GBSA method. In particular, by applying subsequent computational filters, we identified 10 compounds as possible NSP3 Mac1 domain inhibitors. Among them, after the assessment of binding energies (ΔG_bind_) on the whole MD trajectories, we identified the four most interesting compounds that acted as strong binders of the NSP3 Mac1 domain (NSC-358078, NSC-287067, NSC-123472, and NSC-142843), and, remarkably, it could be further characterized for developing innovative antivirals against SARS-CoV-2.

## 1. Introduction

Severe acute respiratory syndrome coronavirus 2 (SARS-CoV-2) has been extensively studied and is known to be the cause of one of the deadliest viral infections worldwide [1,2]. This virus is a single-stranded positive RNA that contains a large genome (30 kb), which contains varying accessory genes, structural and nonstructural proteins known as NSPs [3]. Approximately two-thirds of the SARS-CoV-2 genome is covered by two large open reading frames (ORFs), known as ORF1a and ORF1b, which are located at the 5-end of the genome, and both encode NSPs [4,5]. Translation of these two ORF regions results in the production of two polyproteins, known as PP1a and PP1ab, which are able to auto-cleave themselves and result in the production of varying individual NSP proteins with different functions that have been studied and clarified [6,7,8]. 

NSP proteins can repress the apoptotic pathways in the host cell and prevent the activation of immune signaling pathways that could halt the replication of the virus. NSP3 is a multidomain protein, and each domain carries specific functions [9,10]. As newer studies emerge, the more NSP3 macrodomain of SARS-CoV-2 becomes popular for the design of new antiviral small-molecule inhibitors [10,11]. Mutation experiments have proved that genetic alternations in the NSP3 structure could halt the capability of the virus to replicate [12,13,14]. NSP3 multiprotein contains multiple functional domains within its structure; the ADP ribose phosphatase (ADPR) is its largest domain, and it is also referred to as the macrodomain, x-domain, or Mac1 domain [14,15]. 

The ADPR, or macrodomain, of SARS-CoV shares a high range of sequential similarity (about 80%) with SARS-CoV-2 [16,17]. However, SARS-CoV-2 has better interactions with the ADP-ribose molecule than SARS-CoV. Because of the orientation of key interacting amino acids, such as Phe156, within the adenosine-binding site of the ADPR domain, better H-bonds can be formed, resulting in a better binding affinity, stabilization, and conformational changes of the ADP ribose ligand within this domain [17,18]. ADP ribosylation is a post-translational event that occurs in the host cell and is known as a host-defense mechanism against viral infection, preventing viral replication and activates the immune response [19,20]. 

ADP-ribosyl transferase (ART) proteins are responsible for the covalent addition of ADP-ribose molecules in the form of mono-ADP-ribose (MAR) or poly-ADP-ribose (PAR) tags to proteins and nucleic acids in the presence of NAD^+^ molecules, which are donors of ADP-ribose molecules. ADP ribosylation is a dynamic modification that can be reversed using different classes of proteins that contain macrodomains within their structures. These proteins are capable of detecting and binding to MAR and PAR moieties and are involved in important cellular processes, such as apoptotic and DNA-damage response pathways [21,22]. For instance, poly-ADP-polymerase 14 (PARP14) can use the NAD^+^ molecule and transfer its ADP-ribose moiety to target proteins that can initiate the transcription of antiviral cytokines and prevent viral replication [23,24,25]. Therefore, ADPR is a key domain in the SARS-CoV-2 virus that can protect its replication cycle against immune response signaling pathways of the host [26]. 

Previous research has clarified that other ligands, such as AMP, ADP, and even RNA, can bind to this region [27]. The ADPR domain holds two key sites for the recognition and binding of the ADP-ribose molecule: an adenosine-binding site and a catalytic binding site [28,29]. When it detects ADP-ribose tags on proteins or nucleic acids, it uses water molecules to hydrolyze this covalent modification and release the ADP-ribose tags [30]. Previous studies that attempted to design small-molecule inhibitors have used methods such as fragment-based approaches and molecular docking to introduce fragment structures with high-potential binding sites in the ADPR domain [11,29,31,32,33,34]. However, one problem with their results is the low binding affinity of the fragments with the adenosine binding site of the macrodomain [34], as well as the rigidity of the protein structure and lack of solvent molecules in the molecular docking method [35,36]. 

In this study, molecular docking and molecular dynamics (MD) simulations were used to screen the ligand library and evaluate the docking results in the solved simulation environment. ADMET and physicochemical properties were checked, and an MM/GBSA analysis was performed to further analyze the accuracy of ligand/protein complex binding affinity and stability for each molecule. A detailed illustration of the computational protocol is shown in Figure 1.

## 2. Results

### 2.1. Structure-Based Virtual Screening

According to previous studies, the risk ratio of SARS-CoV-2 infection is higher in patients diagnosed with cancer. Therefore, the National Cancer Institute (NCI) anticancer ligand library was selected for virtual screening against the macrodomain of SARS-CoV-2 NSP3 Mac1 (PDB ID: 6W02) to identify potential inhibitors for further investigations. This library, with over 200,000 selected compounds that were experimentally proven by the Developmental Therapeutic Program (DTP) platform to be effective on multiple human cell lines, was chosen for the docking targeting of the ADP-ribose binding site of NSP3 Mac1. The Molegro Virtual Docker (MVD) software was chosen for this analysis and was set on virtual screening mode to select the top 3% of docked ligands with the best docking scores. MVD software evaluates the binding affinity of docked ligands based on the MolDock and Re-Rank scoring functions, with the latter having better accuracy in predicting binding scores. 

As shown in Table 1, based on Re-Rank scores, NSC-358078 demonstrated the most significant binding score (−193.852), and it had a stronger H-bond interaction energy (−25.305) compared with the rest of the compounds. The compound NSC-353057 also had the best MolDock score (−275.252). However, the Re-Rank scoring function is much more reliable than the MolDock scoring function. Therefore, compounds with better Re-Rank scores, such as NSC-358078 (−193.852), NSC-287067 (−186.814), NSC-353057 (−178.899), and NSC-123472 (−178.751), are predicted to have stronger binding affinities with the binding site targeted in the Mac1 domain than other hits. Moreover, to increase the reliability of the work to provide a final compound selection, we performed 100 ns of MD simulation, considering all the complexes selected by docking studies. Using MD simulation experiments, we calculated the binding energy (ΔG_bind_) of the ligands over the entire MD trajectory, using the MM/GBSA technique.

As shown in Figure 2A, all different domains of the SARS-CoV-2 NSP3 polyprotein are shown, and Figure 2B shows the structural analysis of the ADP-ribose molecule in complex with NSP3 Mac1 (PDB ID: 6W02). As indicated in Figure 2A,B, the key amino acids of the binding site, such as Ile23, Phe156, and Asp22, in the crystal structure were shown to be involved in the formation of conventional H-bonds with the adenine group of ADP-ribose. The distal ribose group, between the adenine and diphosphate groups, interacts with the Leu126 and Ala129 residues. Direct H-bonds were formed between the diphosphate group of ADP-ribose and Val49, Ser128, Gly130, Ile131, and Phe132 residues. There is a rich glycine area within the β3 and α2 loop that interacts with the negatively charged group of ADP-ribose. The glycine-rich region is a particular region of the protein with higher flexibility because of the presence of several glycine residues. The proximal ribose of ADP-ribose interacts and forms H-bonds with this region targeting residue Gly47, in addition to other residues, such as Asn40 and Ala38. 

Figure 3 and Figure 4 show the 2D structures and 2D binding interactions of the top 10 docked hits in complex form with the Mac1 domain of NSP3. According to the binding modes of the compounds shown in the Figures, all the ligands were predicted to form H-bond interactions with residues such as Asp22, Ile23, Ala154, Val49, Asn40, Ile126, and Leu23 and with the residues located in the glycine-rich region area within the β3-α2 loop of the protein, such as Gly46, Gly47, and Gly130. Van der Waals interactions were also commonly detected in the binding of the compounds with residues such as Ala52, Val49, Ile131, Leu160, Leu126, Ile23, Ala38, and Ala50 inside the targeted cavity in the Mac1 domain. Other hydrophobic interactions, such as π-π T-shaped and π-π stacking, were also detected with Phe132 and Phe156 when analyzing the docking output of compounds NSC-729650, NSC-142843, and NSC-338956. The adenyl fragments in compounds NSC-20272 and NSC-358078 were able to interact with Asp22, Ile23, and Phe156 through H-bonds, which also interact with the adenyl group of the ADP-ribose molecule.

The 3D binding conformations for all selected compounds inside the targeted cavity of the Mac1 domain, along with the surface view of the complexes, are provided in Appendix A. Based on the figures, it can be seen that hydrophobic interactions of the ADP ribose binding site inside the Mac1 domain govern the binding mode of the top-ranked compounds. According to the developed computational protocol, the best performing compounds were analyzed for their ADMET properties to understand their role as possible drug candidates. Finally, the stability of the ligand/protein complexes and the ΔG_bind_ of each were assessed by conducting MD simulation studies to select the possible computational hits.

### 2.2. Physicochemical and Drug-like Properties

To estimate the drug-likeness potential of a compound, multiple software and online public databases can be used, which use algorithms that can predict Lipinski’s rule of five, ADMET, and the physicochemical properties of compounds based on their chemical structure. The public SwissADME web server was used to calculate the physicochemical and drug-likeness potentials of the selected docked hits. According to the results reported in Table 2, among the selected hits, NSC-287067 and NSC-142843 had the best physicochemical and drug-like properties, with an appropriate LogP ratio and H-bond donors and acceptors.

### 2.3. MD Simulation and MM/GBSA Analysis

A thorough MD investigation was conducted using Desmond software to improve the accuracy and robustness of the ligand screening (Desmond Molecular Dynamics System, D. E. Shaw Research, New York, NY, USA, 2020; Maestro–Desmond Interoperability Tools, Schrödinger, New York, NY, USA, 2020). MD simulations studies were performed on the ten top-ranked compounds found by the virtual screening procedure, considering ten complexes obtained from molecular docking studies. In this type of ligand-screening method, MD simulation analyses are essential to determine whether a selected molecule can maintain contact with the chosen binding site, whether it might detach from the binding site, or whether the discovered binding mode is unstable during the MD simulation.

As a result, all MD run trajectories were examined to narrow the selection of compounds to those that could successfully interact with the NSP3 Mac1 domain, maintaining the binding to the target protein. Accordingly, the trajectories were assessed by calculating the dynamic ligand-interaction diagrams, root mean square deviation (RMSD), and root mean square fluctuation (RMSF), as well as visually inspecting each trajectory. In general, the NSP3 Mac1 domain in complex with the selected ligands showed a small RMSD value and limited fluctuation events, according to the RMSF calculation, whereas some compounds did not show stable binding mode within the selected binding site (NSC-170987, NSC-338956, and NSC-353057), indicating that they could not properly interact within the selected binding site; therefore, they can be discharged from further evaluation.

To further corroborate the MD simulation results, we calculated the free binding energy (ΔG_bind_), considering the entire MD trajectory of the selected complexes to obtain a clear overview of the binding affinity of the chosen computational hits. Accordingly, by employing the MM/GBSA techniques, we obtained the ΔG_bind_ values that are listed in Table 3. Furthermore, in Table 3, we enclosed the main energy contributions to the overall binding free energy, considering different energy sources. The van der Waals energy contribution, which is one of the free energy components in Table 3, was found to be the main source of the ligand binding energy. This result emphasizes the critical role that hydrophobic interactions play in the stability of ligand/protein complexes, given the hydrophobic nature of the Mac 1 binding site. 

Furthermore, we calculated the ΔG_bind_ considering the entire MD trajectories for the most promising compounds. By conducting an MD analysis, including the ΔG_bind_ calculation, we found the possibility that the four best performing compounds could strongly bind to the NSP3 Mac1 domain with relevant stability and, consequently, should be further investigated for their possible anti-SARS-CoV-2 agents. In the next paragraph, we discuss the most stable complexes (Appendix A), in which the ligands showed the most relevant ΔG_bind_, while the rest of the MD analyses are provided in Appendix A.

#### 2.3.1. MD Simulation Analysis of Compound NSC-358078

One of the most promising compounds selected by the developed computational protocol was the molecule NSC-358078. Figure 5 shows the output of the MD analysis for the complex NSP3 Mac1/NSC-358078. Based on these results, the binding mode of the compound is consistent with that found in molecular docking studies. 

The RMSD and RMSF calculations (Figure 5A–C) showed the high stability of the protein and the ligand, and limited conformational changes were found in the ligand structure, indicating a satisfactory ability of the ligand to target the selected protein, as strongly supported by the ΔG_bind_ value. Furthermore, the compound NSC-358078 maintained its binding with the NSP3 Mac1 binding site, as found by docking studies. The H-bonds were maintained, except for the H-bonds with Asp22 and Gly130, which were no longer detectable after approximately 40 and 10 ns, respectively. The H-bond with Asp22 was replaced by that with Ala21, and regarding the H-bond with Gly130, a more favorable water-mediated H-bond was found with Ala129. Moreover, we detected favorable H-bonds with Ala39, Asn40, Gly46, and Gly47, which contributed to stabilizing the described binding mode (Figure 5D,E). Accordingly, the MD output confirmed that the compound NSC-358078 could strongly bind to the target protein.

#### 2.3.2. MD Simulation Analysis of Compound NSC-287067

Based on the computational evaluation, the compound NSC-287067 showed an interesting profile. Figure 6 illustrates the output of the MD analysis for the complex NSP3 Mac1/NSC-287067. According to the results, the RMSD and RMSF calculations (Figure 6A–C) showed the high stability of the protein and the ligand, and limited conformational changes were found in the ligand structure, indicating a satisfactory ability of the ligand to target the selected protein, maintaining the binding with the NSP3 Mac1 binding site, as found by docking studies. The contacts were maintained, except for hydrophobic contacts with Ile23, Ala38, Ala50, and Ala52, which became sporadic. The H-bond with Asn40 became water-mediated, whereas we detected novel hydrophobic contacts with Leu160. Ile131 was observed to be mainly involved in H-bonding with the compound, and H-bonds with Gly47 and Ser128 became strongly detectable, which contributed to stabilizing the described binding mode (Figure 6D,E). Accordingly, the MD output and the ΔG_bind_ value confirmed that the compound NSC-287067 could act as an NSP3 Mac1 binder.

#### 2.3.3. MD Simulation Analysis of Compound NSC-142843

Again, as observed for the other hits, the RMSD and RMSF calculations (Figure 7A–C) for the compound NSC-142843 showed the high stability of the protein and the ligand, and limited conformational changes were found in the ligand structure. Moreover, although most polar and nonpolar interactions were stable during the simulation time, we observed a lack of the H-bond with Ile23, whereas the H-bonds with Ser128 and Ile131 became sporadic. In contrast, two relevant H-bonds were established by the compound with residues Gly47 and Asp127 (Figure 7D,E). Accordingly, the MD output analysis and the ΔG_bind_ value indicated that the compound NSC-142843 could behave as a ligand of the NSP3 Mac1 domain. 

#### 2.3.4. MD Simulation Analysis of Compound NSC-123472

The last molecule with a favorable profile is compound NSC-123472. The MD analysis showed the high stability of the protein and ligand, and limited conformational changes were found in the ligand structure (Figure 8A–C). The compound maintained the main contacts found in docking studies. In particular, we observed a lack of hydrophobic contacts with Ala50, whereas the hydrophobic contacts with Leu126 became sporadic. In contrast, additional contacts that stabilized the retrieved binding mode were observed. The MD analysis revealed that two relevant water-mediated H-bonds were established by the compound with Lys44 and His45, whereas direct H-bonds were detected with Asp22 and alternatively with Ile23 (Figure 8D,E) and Ala154. Overall, the analysis of the MD output, along with the ΔG_bind_ value, indicated that NSC-123472 could target the NSP3 Mac1 domain.

## 3. Discussion

Since the initial outbreak and emergence of SARS-CoV-2, many genetic variations within its genome have been reported [37], and these were mostly detected in the structure of the viral spike protein and were found to result in a better interaction with the ACE2 receptor of the host cells, increasing the infectivity of the virus [38,39]. However, fewer mutations have been found within the ORF1a and ORF1b regions of SARS-CoV-2, and they encode a variety of NSPs [40,41]. 

The high variation and mutation rate of the SARS-CoV-2 spike protein have created further challenges to and complications for the effectiveness and development of therapeutic antibodies and vaccines [41,42,43]. Therefore, in recent years, more research groups have become interested in the design of new antiviral drugs and small-molecule inhibitors that can target SARS-CoV-2 NSPs [44,45,46]. The translation of the ORF1a and ORF1b regions results in the production of two polyproteins that can cleave themselves with the help of protease domains present in their structure and can lead to the production of 16 varying NSPs with specified functions [2,5]. 

NSP3 is a multidomain protein that carries multiple domains within itself. Its largest domain is known as the Mac1 domain, or ADRP [11,12,17]. Previous studies have shown that the NSP3 Mac1 domain is capable of detecting and binding with ADP-ribose moieties that are attached to proteins or nucleic acids and can remove them in the presence of water molecules [27]. ADP-ribosylation is a post-translation covalent modification applied to proteins with the help of ADP-ribosyltransferase (ART) enzymes, which belong to the host cells and are responsible for tagging proteins with ADP-ribose moieties in a timely manner during viral infection, resulting in the regulation of DNA damage and immune responses [22]. The NSP3 Mac1 domain has been suggested by previous studies to be capable of interfering with the interferon response by removing the mono-ADP-ribose tags from specific host proteins [41]. 

Mutations in the Mac1 domain can result in decreased enzymatic capability in the removal of ADP-ribose tags and increased viral variability in interferon signaling [38,41]. In 2020, Claverie et al. hypothesized that STAT1 is a potential target of the SARS-CoV-2 NSP3 Mac1 domain of de-mono-ADP-ribosylation, which could lead to increased phosphorylated levels of STAT1 and higher expression levels of ACE2 that can result in increased viral infection [47]. A similar conformation has also been found in the ADP-ribose binding site of PARP14 and the Mac1 domain of SARS-CoV-2 [26,47]. However, the SARS-CoV-2 NSP3 Mac1 domain reverses the modifications performed by PARP14 by hydrolyzing the mono-ADP-ribose moieties [24,26]. The PARG protein is a member of the macrodomain family that has a function similar to that of the SARS-CoV-2 NSP3 Mac1 domain and is capable of recognizing and hydrolyzing ADP-ribose tags at DNA damage sites and initiating a DNA repair response [48]. In 2021, a study performed a virtual screening analysis on the SARS-CoV-2 NSP3 Mac1 domain, using a library of compounds that were previously identified as PARG inhibitors [32]. 

In 2021, another study introduced new inhibitors against the SARS-CoV-2 NSP3 Mac1 domain via a virtual screening analysis on ZINC15 drug-like libraries [49]. A crystal structure analysis of the SARS-CoV-2 NSP3 Mac1 domain by Michalska et al. divided the ADP-ribose binding cavity of Mac1 into adenosine-binding and catalytic sites [27]. In 2021, Schuller et al. used multiple fragment libraries against the same site in the SARS-CoV-2 NSP3 Mac1 domain. In addition, their study showed that amino acids such as Asp22 and Phe156 could help the adenosine site reach a higher conformational change range in the SARS-CoV-2 NSP3 Mac1 domain [34]. However, one problem with the fragment-based approach is the low binding affinity of fragments with the adenosine-binding site in the Mac1 domain. 

A multicenter study performed by Dai et al. revealed that COVID-19 infection correlated with greater risks and more severe outcomes in patients who were also diagnosed with cancer [50]. It was also suggested that patients with cancers, such as lung cancer and metastatic cancers, are more susceptible to SARS-CoV-2 infection. Therefore, in this study, we used the NCI anticancer library with over 200,000 synthetic and natural-derived compounds that have been filtered by the DPT research program through a series of cell-based experiments on a variety of human cancer cell lines. This library was selected for a virtual screening campaign against the ADP-ribose binding site of the SARS-CoV-2 NSP3 Mac1 domain. The top 3% of the screened compounds were selected on the basis of their MolDock and Re-Rank scores. 

The crystal structure of the Mac1 domain revealed the significant involvement of key amino acids, namely Ile23, Phe156, and Asp22, in the formation of conventional H-bonds with the adenine group of ADP-ribose. Notably, the distal ribose group situated between the adenine and diphosphate moieties was found to interact with Leu126 and Ala129. In addition, direct H-bonds established connections between the diphosphate group of ADP-ribose and Val49, Ser128, Gly130, Ile131, and Phe132 residues. Within the β3 and α2 loops, a region rich in glycine serves as an interactive site for the negatively charged group of ADP-ribose. Moreover, the proximal ribose of ADP-ribose establishes H-bonds with the glycine-rich sequence, encompassing residues Gly47, Asn40, and Ala38.

Based on the assessment of the physicochemical and ADMET properties of the top selected hits, some selected compounds showed suitable drug-like features. We then performed MD simulation experiments on the top 10 selected NSC compounds in complex with the SARS-CoV-2 NSP3 Mac1 domain. For some of them, we detected significant stability inside the ADP-ribose binding site of the Mac1 domain. Finally, the ΔG_bind_ values of the compounds were calculated using the MM/GBSA technique. Accordingly, on the basis of the developed computational protocol, we identified four promising molecules that could strongly target the SARS-CoV-2 NSP3 Mac1 domain. Based on the computational receptor-based assessment, including the calculation of ΔG_bind_ and the favorable drug-like profile, we identified the following compounds as potential SARS-CoV-2 NSP3 Mac1 inhibitors: NSC-358078, NSC-287067, NSC-123472, and NSC-142843. Our study can inspire future studies and global investigations on the drug-like potential of these top selected NSC compounds against a variety of human cancer cell lines and their response to SARS-CoV-2 infection.

## 4. Materials and Methods

### 4.1. Selection of the Ligand Library

Because patients with cancer are at a higher risk of SARS-CoV-2 infection, we selected the anticancer ligand library of NCI [51,52] to identify novel anticancer compounds that can also halt the activity of SARS-CoV-2. This library includes approximately 200,000 ligands with a combination of natural and synthetic compounds that were evaluated in two human cancer cell line experiments with varying doses by DTP (https://dtp.cancer.gov, accessed on 25 February 2023). DTP works as part of the NCI research program for the development of novel and effective anticancer compounds.

### 4.2. Structural Preparation of Protein and Ligands

Before performing molecular docking screening, it is vital to optimize the structure of the protein of interest and the selected library of ligands into their best optimum conformation. The structural optimization of compounds into their energy-minimized 3D conformation results in enhanced binding pose and interaction of ligands with amino acid residues in the active site of the target protein. As represented in the flowchart in Figure 1, the next step after the selection of a ligand library is the structural preparation of the ligands and protein. The LigPrep application (LigPrep, Schrödinger, LLC, New York, NY, USA, 2020), a module of the Schrodinger Suite 2020 package (Schrödinger, LLC, New York, NY, USA, 2020), was used for energy minimization and 3D structural optimization of the compounds, along with enhancing their ionization state at the cellular pH level (7.4 ± 0.5). 

The RCSB database (https://www.rcsb.org, accessed on 15 February 2023) is a free public database that contains thousands of 3D crystal structures of proteins. The structure of the SARS-CoV-2 NSP3 Mac1 was downloaded from the RCSB databank (PDB ID: 6W02) and then prepared using the Protein Preparation Wizard (PPW) module of the Schrödinger Suite 2020 package. The PPW includes multiple steps, including the addition of hydrogen atoms and disulfide bonds that were missing in the initial structure, removal of water and cofactor molecules, modification of amino acid’s protonation state, and energy minimization that enhances the conformation of protein for better performance during molecular docking analysis [53]. The OPLS3 force field was selected, and the RMSD cutoff value was set as 0.30 Å. 

### 4.3. Receptor Grid Generation

The selection of a specific grid box on the ligand binding site in the protein’s structure is an initial step before conducting molecular docking analysis [54]. The grid box is basically a 3D region that covers the area around the protein’s active site and includes all kinds of potential energies from the interaction of the ligand with the amino acids in the target ligand binding site of the protein. Based on the goal of the current study, a grid box was set at the binding site of the ADP-ribose molecule with chain A of NSP3 Mac1, which is surrounded by multiple loops formed between alpha helixes and beta strands. MVD software (version 7.0.0) was used to generate a grid box on the interest domain of the NSP3 Mac1 with the following coordination: x = 3.69 Å, y = −6.11 Å, and z = −22.36 Å.

### 4.4. Structure-Based Virtual Screening

The molecular docking analysis is a technique that is widely used for screening large libraries of compounds with varying conformational binding poses with the target site of the protein’s crystal structure [55,56,57,58]. Virtual screening is a calculation module for the molecular docking analysis that selects the top percentage of docked compounds based on their best docking scores. The prepared NCI anticancer library [51,52] was used for a molecular docking analysis against NSP3 Mac1, using MVD software [59], and the virtual screening module was set on the top 3% of the screened compounds with significant binding scores [60]. The MVD software uses two scoring functions, MolDock and Re-Rank. 

The MolDock or Grid scoring function [61] is based on the THOMSEN 2006 scoring function, which orders the best-docked compounds based on the grid box located on the protein’s active site. The Re-Rank scoring function has better prediction potential than the MolDock scoring function. It calculates the binding affinity of docked compounds by summing the internal energy of the ligand and all the interaction energies between the ligand and protein. The number of docking runs was set to 20 to repeat the docking run of each ligand 20 times, resulting in more reliable docking scores. In addition, H-bond optimization was chosen to enhance the direction of H-bonds within the ligand/protein complex. The energy minimization option was also chosen after the docking screening analysis. Discovery Studio software (version 4.5, 2021, Dassault Systemes BIOVIA, 78140 Vélizy-Villacoublay, Francia) was used to visualize different types of interactions.

### 4.5. Analysis of ADMET and Physicochemical Properties

Lipinski’s rule of five is a general estimation of the physicochemical properties of compounds that considers multiple factors to predict compounds with drug-like potential [62,63,64]. These descriptors—along with the other physicochemical properties, such as the LogS ratio, TPSA, and bioavailability score, of the top docked compounds—were all calculated using the SwissADME (http://www.swissadme.ch, accessed on 20 March 2023) online public database [65,66].

### 4.6. MD Simulation and MM/GBSA Calculation

In the previous step, a molecular docking analysis was performed as a practical tool for screening large libraries of ligands and finding ligands that are predicted to have a significant affinity for binding with the target site in the NSP3 Mac1 domain. However, molecular docking has its own disadvantages with respect to accuracy. It considers protein as a rigid structure in a vacuum environment in which its conformation is kept rigid at its energy-minimized state. Previous theories suggest that the old lock-and-key theory does not apply anymore to clearly explain the binding process of the ligand with its targeted binding site. Therefore, MD simulation analysis can be used as a tool for validating molecular docking results because of its various advantages [67,68]. During an MD simulation, the structures of proteins and ligands are both given conformational freedom to move and adapt to their best conformation in the solved state with an appropriate concentration of ions at equilibrium conditions [69]. By simulating the movement and interactions of atoms and molecules in a system over time, researchers can gain insights into important processes [70]. Before starting the simulation, each complex derived from docking studies was imported into Maestro, providing a graphical interface for the Desmond software to set the MD simulation parameters (Desmond 6.4 academic version, provided by D. E. Shaw Research (“DESRES”), Desmond Molecular Dynamics System, D. E. Shaw Research, New York, NY, USA, 2020. Maestro–Desmond Interoperability Tools, Schrödinger, LLC, New York, NY, USA, 2020). The complex is then filled and solvated into a cubic box containing water molecules, using the TIP3P model. The TIP3P model is known for its accuracy in describing water interactions [71,72]. A Md simulation was conducted using the OPLS force field [73]. The simulation was computed using two NVIDIA GPUs, which provided the necessary speed and accuracy. To simulate the physiological concentration of monovalent ions, Na^+^ and Cl^−^ ions were added to achieve a final salt concentration of 0.15 M. The MD simulations were performed under NPT (constant number of particles, pressure, and temperature), as an ensemble class, with a constant temperature of 300 K and pressure of 1.01325 bar. The equations that estimate motion were calculated using the RESPA integrator, with an inner time step of 2.0 fs for interactions such as bonded and non-bonded interactions within the short-range cutoff [74]. Nose–Hoover thermostats were also applied to keep the temperature constant within the simulation [75], while the Martyna–Tobias–Klein method was used to control the pressure [76]. The cutoff for van der Waals and short-range electrostatic interactions was set at 9.0 Å, and long-range electrostatic interactions were calculated using the particle-mesh Ewald method (PME) [77]. To ensure that the system was equilibrated, a default protocol that includes a series of restrained minimizations and MD simulations was used to gradually relax the system. In addition, the ADP ribose molecule was considered to be the reference ligand in the simulation to obtain a direct comparison. The trajectories produced during the MD simulation were analyzed using the Simulation Event option provided in the Desmond package. RMSD_x_ is defined as the average distance between the positions of the corresponding atoms in the two structures. The calculation of RMSD_x_ involves computing the average of the distances between the corresponding atoms in the target and reference frames after superimposing the atoms in the target onto the reference frame. This method is used to measure the deviation of the target frame from the reference frame. The RMSD values were calculated according to the following equation: $${RMSD}_{x}=\sqrt{\frac{1}{N}\sum_{i=1}^{N}{\left({r\prime }_{i}\left(t_{x}\right)-r_{i}(t_{ref})\right)}^{2}}$$
where RMSD_x_ is used to represent the calculation for a frame x, N shows the number of atoms in the system, t_ref_ is used to demonstrate the reference time, and r’ represents the location of the chosen atoms in frame x after superimposing their position based on the reference frame, where frame x is recorded at time t_x_. Proteins are considered to be the building blocks of the human body; each protein has special and important functions, but many of them are still unknown. Every protein has a unique structure, which is necessary for its proper functioning. An essential metric used to quantify the structural dynamics of proteins is the rRMSF. The RMSF is calculated by measuring the deviation of the position of an atom or residue from its average position over an atomic trajectory. RMSF values were also calculated based on the following equation:$${RMSF}_{i}=\sqrt{\frac{1}{T}\sum_{t=1}^{T}<{\left({r\prime }_{i}\left(t\right)-r_{i}(t_{ref})\right)}^{2}>}$$

RMSF_i_ represents the generic residue i, T is referred to trajectory time and the period which the RMSF is calculated, the reference time is shown by t_ref_, r_i_ represents the location of the residue i; the exact position of atoms in the residue i after being superposed based on the reference is demonstrated by r′, and the angle brackets represent the average of the square distance, which is taken over based on the selection of atoms in the residue.

The thermal MM/GBSA script available in Desmond (thermal_mmgbsa.py) [76] was used to evaluate the ΔG_bind_ for the selected complexes. This tool used the Desmond MD trajectory, splitting it into individual frame snapshots, and ran each one through MM/GBSA analysis. During the MM/GBSA calculation, 1000 snapshots from the 100 ns MD simulation were used as inputs to compute the average binding free energy. The evaluated ΔG_bind_ scores are reported as average values in the Results and Discussion section, along with the energy components used in the calculation.

## 5. Conclusions

In summary, we developed a virtual screening protocol, using the NCI anticancer library, considering the SARS-CoV-2 NSP3 Mac1 domain as the drug target. The top-scored hits, with the help of MVD, which uses MolDock and Re-Rank scoring functions for the evaluation of ligand binding affinities, were selected for further analysis. A physicochemical profile evaluation of the top hits revealed satisfactory drug-like potential for some of them. The binding stability and affinity of top NSC compounds against the SARS-CoV-2 NSP3 Mac1 enzyme were assessed by performing MD simulation studies, and their interaction energies were evaluated by calculating the ΔG_bind_ on the whole trajectories. Following this procedure, we identified four drug-like computational hits that could significantly act as NSP3 Mac1 inhibitors that should be further characterized to evaluate their potential antiviral activity against SARS-CoV-2. 

## Figures and Tables

**Figure 1 viruses-15-02291-f001:**
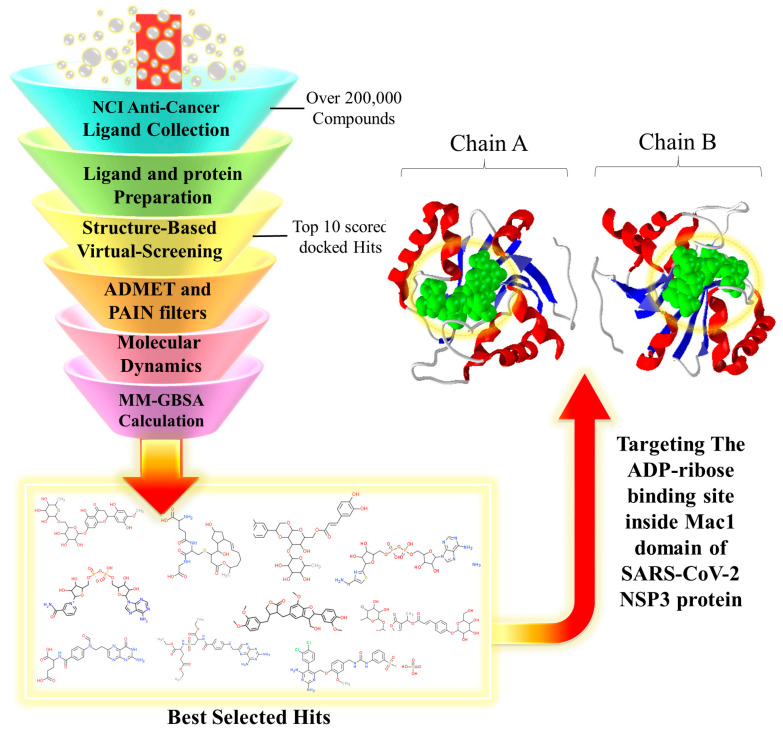
Schematic of the steps followed in the current study for the virtual screening process to identify novel SARS-CoV-2 NSP3 Mac1 inhibitors. Top filtered hits from the NCI anticancer library were selected on the basis of their docking score and interaction energies with the ADP-ribose binding site inside the Mac1 domain. The initial ADP-ribose molecules that were crystallized with both chains of the Mac1 domain are represented in green.

**Figure 2 viruses-15-02291-f002:**
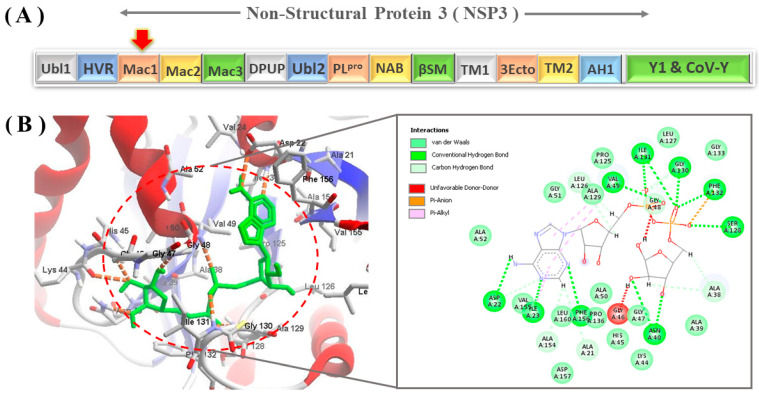
(**A**) All 16 varying domains of SARS-CoV-2 NSP3 Mac1 are illustrated. The Mac1 domain is located at the 5’ end of the NSP3 genome. (**B**) Representation of the ADP-ribose binding conformation and its interaction with key amino acids inside the Mac1 domain.

**Figure 3 viruses-15-02291-f003:**
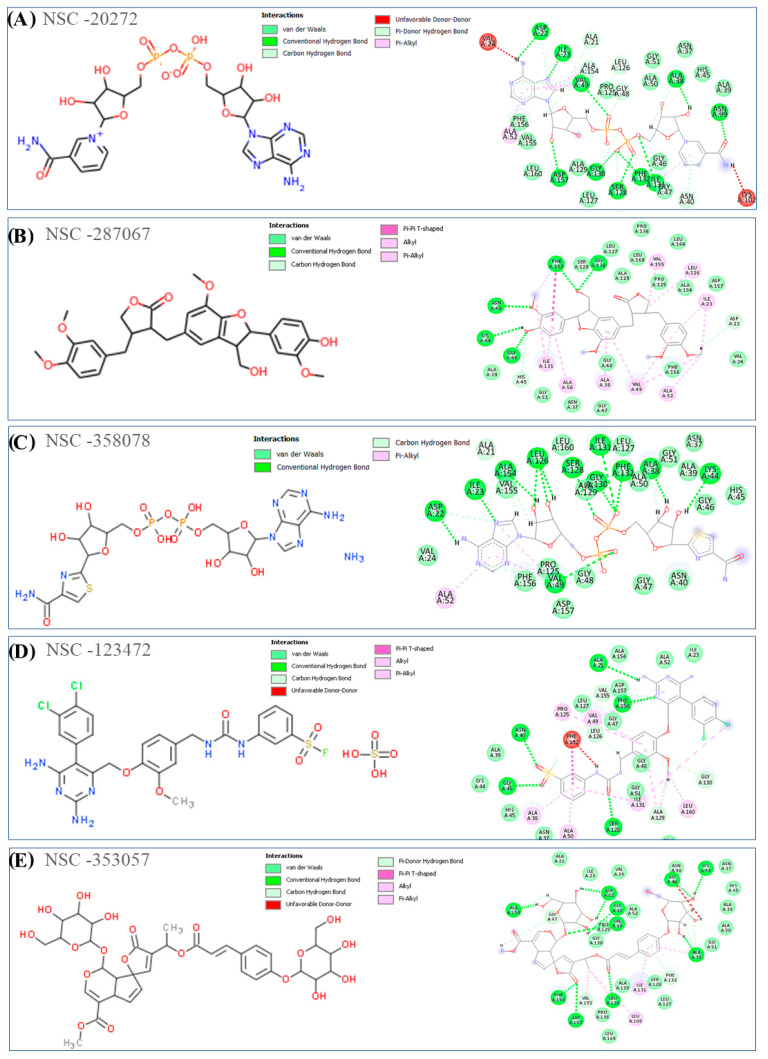
(**A**–**E**) The 2D structures of the selected NSC hits from the virtual screening analysis of the NCI anticancer library are shown, along with their key interactions with amino acids inside the NSP3 Mac1 domain, as determined by docking studies, using MVD software. Discovery Studio software (2021) was used to visualize different types of interactions, such as H-bond, electrostatic, and steric interactions.

**Figure 4 viruses-15-02291-f004:**
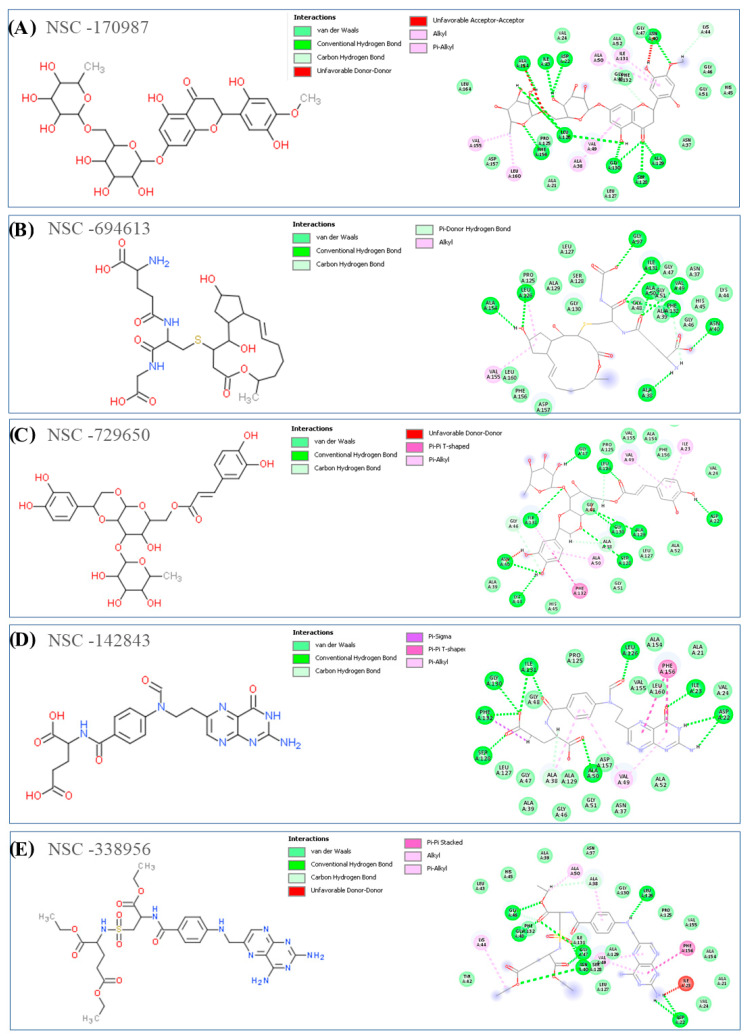
(**A–E**) The 2D structures of the selected NSC hits from virtual screening analysis of the NCI anticancer library are shown, along with their key interactions with amino acids inside the NSP3 Mac1 domain, as determined by docking studies, using MVD software. Discovery Studio software (2021) was used to visualize different types of interactions, such as H-bond, electrostatic, and steric interactions.

**Figure 5 viruses-15-02291-f005:**
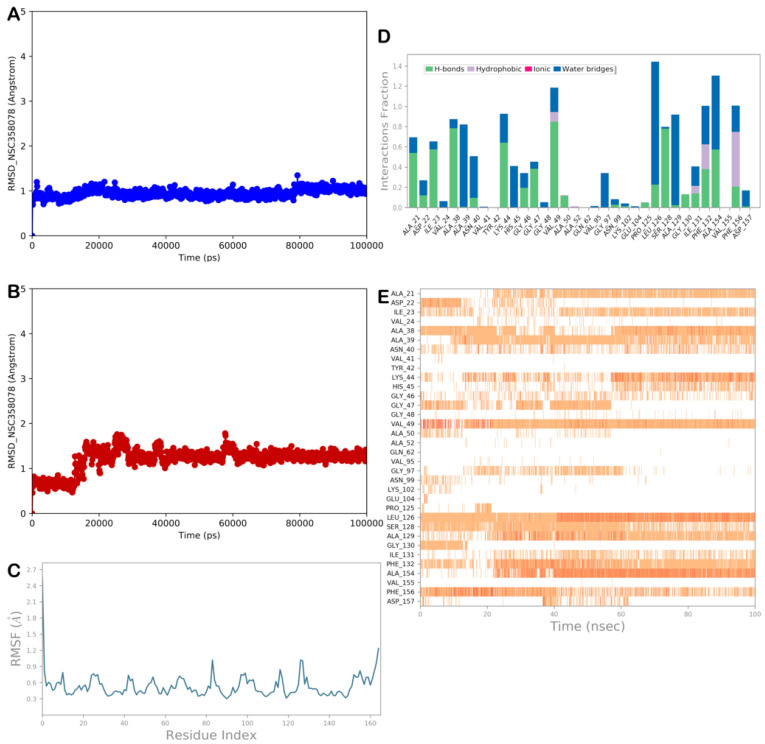
MD analysis of compound NSC-358078. (**A**,**B**) RMSD of the protein (blue line) and of the ligand (red line). (**C**) RMSF analysis of the protein. (**D**,**E**) NSC-358078 monitored during the MD run. The interactions can be grouped into four types: H-bonds (green), hydrophobic (gray), ionic (magenta), and water bridges (blue). The stacked bar charts are normalized over the course of the trajectory: for example, a value of 0.7 suggests that, for 70% of the simulation time, the specific interaction is maintained. Values over 1.0 are possible, as some protein residue may make multiple contacts of the same subtype with the ligand. The subsequent diagram in the figure illustrates a timeline description of the main interactions. The output shows which residues interact with the ligand in each trajectory frame. Some residues make more than one specific contact with the ligand, which is represented by a darker shade of orange (Maestro, Schrödinger LLC, release 2020-3).

**Figure 6 viruses-15-02291-f006:**
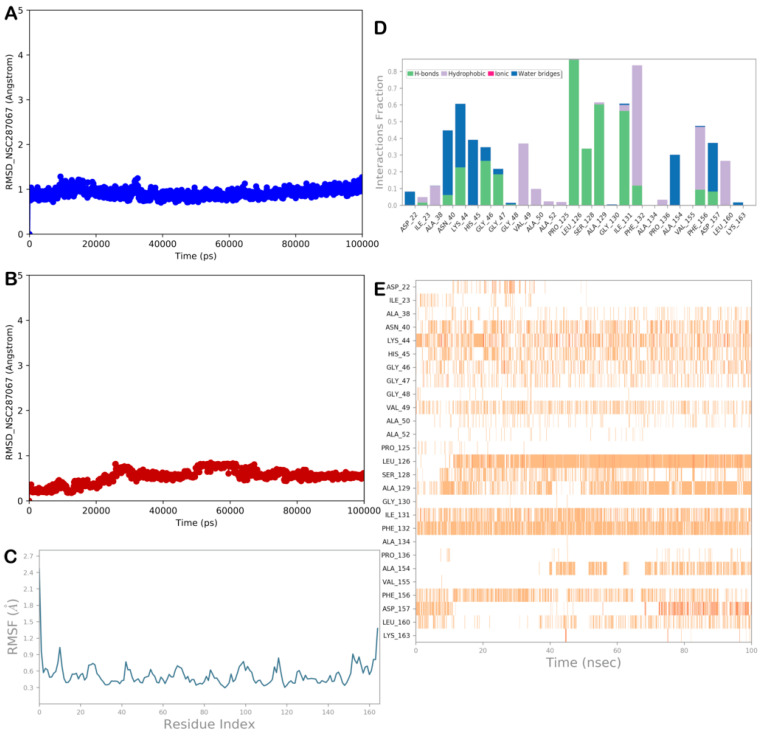
MD analysis of compound NSC-287067. (**A**,**B**) RMSD of the protein (blue line) and of the ligand (red line). (**C**) RMSF analysis of the protein. (**D**,**E**) NSC-287067 monitored during the MD run. The interactions can be grouped into four types: H-bonds (green), hydrophobic (gray), ionic (magenta), and water bridges (blue). The stacked bar charts are normalized over the course of the trajectory: for example, a value of 0.7 suggests that, for 70% of the simulation time, the specific interaction is maintained. Values over 1.0 are possible, as some protein residue may make multiple contacts of the same subtype with the ligand. The subsequent diagram in the figure illustrates a timeline description of the main interactions. The output shows which residues interact with the ligand in each trajectory frame. Some residues make more than one specific contact with the ligand, which is represented by a darker shade of orange (Maestro, Schrödinger LLC, release 2020-3).

**Figure 7 viruses-15-02291-f007:**
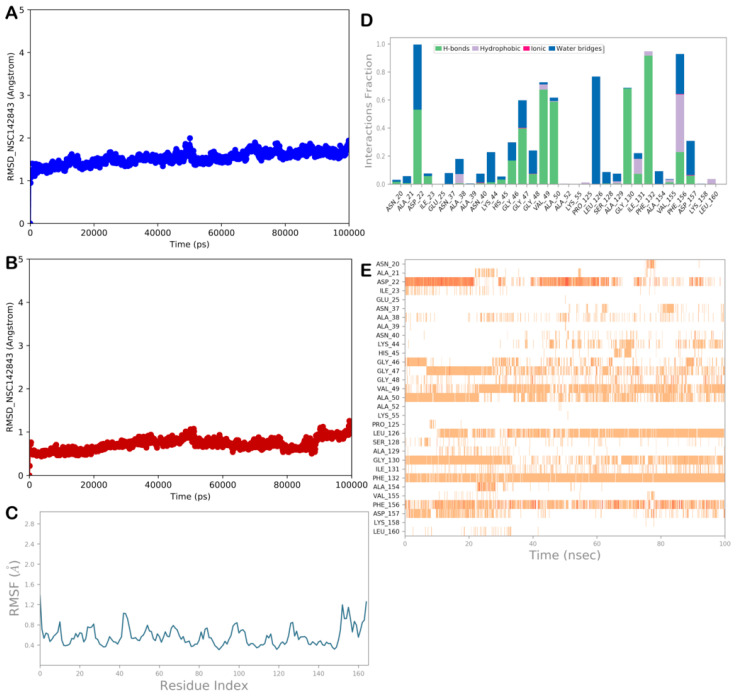
MD analysis of compound NSC-142843. (**A**,**B**) RMSD of the protein (blue line) and of the ligand (red line). (**C**) RMSF analysis of the protein. (**D**,**E**) NSC-142843 monitored during the MD run. The interactions can be grouped into four types: H-bonds (green), hydrophobic (gray), ionic (magenta), and water bridges (blue). The stacked bar charts are normalized over the course of the trajectory: for example, a value of 0.7 suggests that, for 70% of the simulation time, the specific interaction is maintained. Values over 1.0 are possible, as some protein residues may make multiple contacts of the same subtype with the ligand. The subsequent diagram in the figure illustrates a timeline description of the main interactions. The output shows which residues interact with the ligand in each trajectory frame. Some residues make more than one specific contact with the ligand, which is represented by a darker shade of orange (Maestro, Schrödinger LLC, release 2020-3).

**Figure 8 viruses-15-02291-f008:**
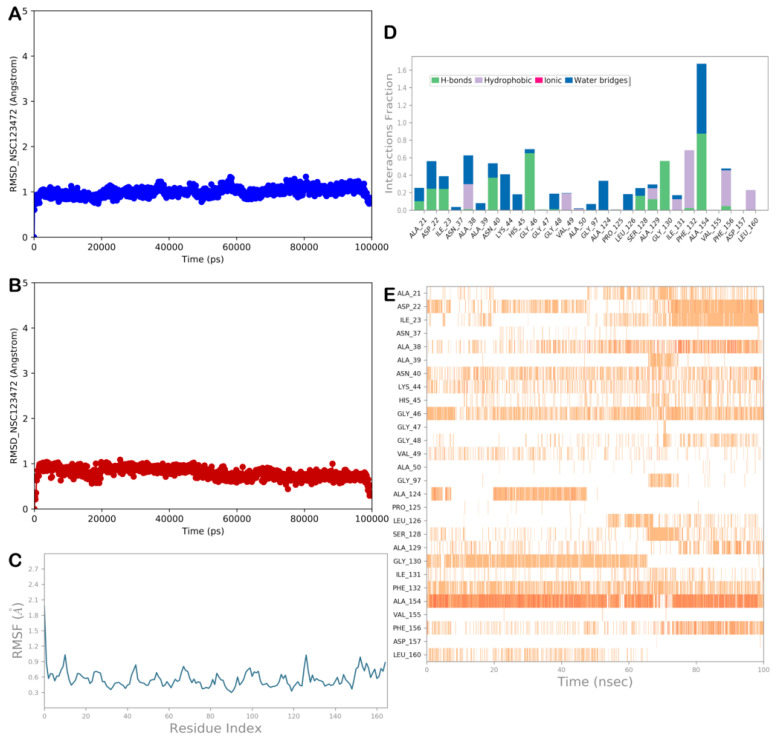
MD analysis of compound NSC-123472. (**A**,**B**) RMSD of the protein (blue line) and of the ligand (red line). (**C**) RMSF analysis of the protein. (**D**,**E**) NSC-123472 monitored during the MD run. The interactions can be grouped into four types: H bonds (green), hydrophobic (gray), ionic (magenta), and water bridges (blue). The stacked bar charts are normalized over the course of the trajectory: for example, a value of 0.7 suggests that, for 70% of the simulation time, the specific interaction is maintained. Values over 1.0 are possible, as some protein residues may make multiple contacts of the same subtype with the ligand. The subsequent diagram in the figure illustrates a timeline description of the main interactions. The output shows which residues interact with the ligand in each trajectory frame. Some residues make more than one specific contact with the ligand, which is represented by a darker shade of orange (Maestro, Schrödinger LLC, release 2020-3).

**Table 1 viruses-15-02291-t001:** Virtual screening results of the NCI anticancer library against SARS-CoV-2 NSP3 Mac1.

Compound ID	MolDock Score	Re-Rank Score	H-Bond
**NSC-358078**	−262.398	−193.852	−25.305
**NSC-287067**	−242.883	−186.814	−9.8173
**NSC-353057**	−275.252	−178.899	−20.004
**NSC-123472**	−236.761	−178.751	−5.5102
**NSC-20272**	−241.454	−176.795	−21.878
**NSC-170987**	−252.206	−175.426	−21.41
**NSC-694613**	−256.587	−174.499	−16.771
**NSC-729650**	−227.151	−174.234	−16.103
**NSC-142843**	−214.877	−172.67	−20.962
**NSC-338956**	−231.941	−169.875	−13.263

**Table 2 viruses-15-02291-t002:** Physicochemical and drug-like properties of the top 10 selected hits from docking analysis.

Compound ID	H-Bond Donor	H-Bond Acceptor	Molecular Weight (g/mol)	LogP	LogS	TPSA (Å^2^)	Bioavailability Score
NSC-358078	9	19	686.48 g/mol	−4.57	1.96	378.37	0.11
NSC-287067	2	9	550.60 g/mol	3.73	−5.57	112.91	0.55
NSC-353057	8	19	778.71 g/mol	−1.47	−3.1	286.89	0.11
NSC-123472	6	12	719.55 g/mol	3.29	−5.7	262.91	0.17
NSC-20272	7	17	663.43 g/mol	−4.87	0.25	340.71	0.11
NSC-170987	9	16	626.56 g/mol	−1.06	−3.16	254.52	0.17
NSC-694613	7	11	587.68 g/mol	−0.9	−0.96	250.88	0.11
NSC-729650	8	15	622.57 g/mol	−0.53	−2.98	234.29	0.17
NSC-142843	5	10	483.43 g/mol	−0.22	−1.61	221.56	0.11
NSC-338956	5	14	675.71 g/mol	0.85	−3.07	278.18	0.17

**Table 3 viruses-15-02291-t003:** Predicted binding free energies (ΔG_bind_), as determined by MM/GBSA calculations, and the energy components of the best-ranked compounds.

Compound ID	ΔG_vdw_ ^a^ (kcal/mol)	ΔG_coul_ ^b^ (kcal/mol)	ΔG_Hbond_ ^c^ (kcal/mol)	ΔG_Lipo_ ^d^ (kcal/mol)	ΔG_Pack_ ^e^ (kcal/mol)	ΔG_SolGB_ ^f^ (kcal/mol)	ΔG_bind_ ^g^ (kcal/mol)
NSC-358078	−71.89	−37.66	−7.09	−19.83	−1.69	35.11	−89.23
NSC-287067	−67.12	−26.19	−3.12	−37.76	−1.35	28.83	−86.75
NSC-142843	−66.43	−32.83	−5.58	−21.48	−1.22	34.03	−83.74
NSC-123472	−64.11	−30.91	−3.91	−24.59	−0.98	31.27	−81.92
NSC-170987	−61.51	−31.37	−5.71	−23.38	−1.10	33.94	−77.31
NSC-20272	−63.79	120.58	−8.16	−22.15	−1.88	−123.53	−76.85
NSC-729650	−55.92	−22.63	−3.49	−38.06	−1.49	31.78	−69.46
NSC-338956	−60.89	−32.86	−5.01	−16.17	−0.64	39.29	−63.56
NSC-353057	−45.14	−34.90	−4.19	−31.89	−0.43	45.36	−54.57
NSC-694613	−40.74	−13.02	−4.27	−20.25	0.00	19.95	−48.87

^a^ Contribution of van der Waals interaction energy to binding free energy. ^b^ Contribution of Coulomb energy to binding free energy. ^c^ H-bonding contributions to binding free energy. ^d^ Lipophilic energy contribution to binding free energy. ^e^ π−π packing energy contribution to binding free energy. ^f^ Generalized Born electrostatic solvation energy contribution to binding free energy. ^g^ Total binding free energy.

## Data Availability

Data are contained within the article and Appendix A.

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
