# Peer review of "Structure-Based High-Throughput Virtual Screening and Molecular Dynamics Simulation for the Discovery of Novel SARS-CoV-2 NSP3 Mac1 Domain Inhibitors"

_viruses, 2023, doi:10.3390/v15122291_

Round 1

Reviewer 1 Report

Comments and Suggestions for Authors

The manuscript by Yazdani, B. et al entitled “Structure-based High-throughput Virtual Screening and Molecular Dynamics Simulation for the Discovery of Novel SARS-3 CoV-2 NSP3 Mac1 domain Inhibitors” conducted a study to identify potential inhibitors from anticarcinogenic library for the NSP3 Mac1 domain of the SARS-CoV-2 virus. Identifying such inhibitors is important for development of drugs for treatment of oncologic patients suffering from coronavirus disease inflicted by severe acute respiratory syndrome coronavirus 2. The researchers performed a virtual screening campaign using the NCI anticancer library and computationally evaluated the library against the NSP3 Mac1 domain. The advantage against the previous studies is that the authors combine docking with molecular dynamics while they leave conformational freedom to the both proteins and ligands, also within the presence of explicit watery media. By using this approach, the Authors identify four drug-like computational hits that could significantly act as NSP3 Mac1 inhibitors that should be further characterized to evaluate their potential antiviral activity against SARS-CoV-2, as supported also by means of physicochemical and ADMET analysis.

The manuscript reads well. The work brings original results obtained by combining state-of-the-art and leading-edge computational tools, software and up-to-date databases. The acquired knowledge is significant for development of drugs for patients with COVID combined comorbidities. The presentation of the results is good. The study investigates possible inhibitors for NSP3 domain, that could replication and transcription of SARS-CoV-2 from insides of its lifecycle. This provides scientifically sound alternatives for stopping the viral infection. As such, I believe the work has a potential to attract interest of wider spectrum of readers across disciplines.

I would have only minor comments.

First of all, panels on Figure 5 and 6 show evolution of contacts along the trajectory. How are these contacts evaluated? I believe the contacts, such as hydrogen bonds in the molecular mechanistic force-field are treated as a part of van der Waals and Coulombic forces. How a hydrogen bond is identified? Could the authors provide also life-time of the interactions? I believe this would provide important information on the stability of the complexes along with the Gibbs free energy.

Secondly, the large blocks of text in several sections, such as Introduction, Discussion and Materials and Methods (4.6) could be divided into smaller paragraphs to improve reading experience.

After these minor questions are answered, the manuscript has my recommendation to be published in the journal.

Author Response

Reviewer 1

The manuscript by Yazdani, B. et al entitled “Structure-based High-throughput Virtual Screening and Molecular Dynamics Simulation for the Discovery of Novel SARS-3 CoV-2 NSP3 Mac1 domain Inhibitors” conducted a study to identify potential inhibitors from anticarcinogenic library for the NSP3 Mac1 domain of the SARS-CoV-2 virus. Identifying such inhibitors is important for development of drugs for treatment of oncologic patients suffering from coronavirus disease inflicted by severe acute respiratory syndrome coronavirus 2. The researchers performed a virtual screening campaign using the NCI anticancer library and computationally evaluated the library against the NSP3 Mac1 domain. The advantage against the previous studies is that the authors combine docking with molecular dynamics while they leave conformational freedom to the both proteins and ligands, also within the presence of explicit watery media. By using this approach, the Authors identify four drug-like computational hits that could significantly act as NSP3 Mac1 inhibitors that should be further characterized to evaluate their potential antiviral activity against SARS-CoV-2, as supported also by means of physicochemical and ADMET analysis.

The manuscript reads well. The work brings original results obtained by combining state-of-the-art and leading-edge computational tools, software and up-to-date databases. The acquired knowledge is significant for development of drugs for patients with COVID combined comorbidities. The presentation of the results is good. The study investigates possible inhibitors for NSP3 domain, that could replication and transcription of SARS-CoV-2 from insides of its lifecycle. This provides scientifically sound alternatives for stopping the viral infection. As such, I believe the work has a potential to attract interest of wider spectrum of readers across disciplines.

Authors: We sincerely thank the reviewer for the positive evaluation of our manuscript.

I would have only minor comments.

First of all, panels on Figure 5 and 6 show evolution of contacts along the trajectory. How are these contacts evaluated? I believe the contacts, such as hydrogen bonds in the molecular mechanistic force-field are treated as a part of van der Waals and Coulombic forces. How a hydrogen bond is identified? Could the authors provide also life-time of the interactions? I believe this would provide important information on the stability of the complexes along with the Gibbs free energy.

Authors: We apologize for the lack of details regarding the evaluation of the number and quality of contacts established by the ligand within the selected binding site during the MD simulation. The types of interactions (Figures 5-8 panel D)can be categorized by type and are summarized as follows: hydrogen bonds, hydrophobic, ionic, and water bridges. Each interaction type contains more specific subtypes, which can be explored through the Simulation Interactions Diagram panel. Stacked bar charts are normalized over the course of the trajectory. For example, a value of 0.7 suggests that the specific interaction is maintained for 70% of the simulation time. Values over 1.0 are possible because some protein residues may make multiple contacts of the same subtype with the ligand. Hydrogen bonds: (H-bonds) can be further broken down into four subtypes: backbone acceptor, backbone donor, side-chain acceptor; side-chain donor. The current geometric criteria for protein-ligand H-bonds are a distance of 2.5 Å between the donor and acceptor atoms (D—H···A); a donor angle of ≥120° between the donor–hydrogen– acceptor atoms (D—H···A); and an acceptor angle of ≥90° between the hydrogen-acceptor-bonded_atom atoms (H···A—X). Hydrophobic contacts: fall into three subtypes: p-cation, p-p; and other, non-specific interactions. Generally, these types of interactions involve a hydrophobic amino acid and an aromatic or aliphatic group on the ligand; however, we have extended this category to include p-cation interactions. The current geometric criteria for hydrophobic interactions are as follows: p-cation — Aromatic and charged groups within 4.5 Å; p-p — Two aromatic groups stacked face-to-face or face-to-edge; Other — A non-specific hydrophobic sidechain within 3.6 Å of a ligand’s aromatic or aliphatic carbons. Ionic interactions, or polar interactions, are between two oppositely charged atoms that are within 3.7 Å of each other and do not involve a hydrogen bond. Water bridges are hydrogen-bonded protein– ligand interactions mediated by water molecules. The hydrogen-bond geometry is slightly relaxed from the standard H-bond definition. The current geometric criteria for protein– water or water-ligand H-bond areas are as follows: a distance of 2.8 Å between the donor and acceptor atoms (D—H···A); a donor angle of ≥110° between the donor–hydrogen– acceptor atoms (D—H···A); and an acceptor angle of ≥90° between the hydrogen-acceptor-bonded_atom atoms (H···A—X).

Regarding the timeline interactions (Figures 5-8 panel E), the output shows which residues interact with the ligand in each trajectory frame. Some residues make more than one specific contact with the ligand, which is represented by a darker shade of orange, according to the scale to the right of the plot.

The explanation for the type of contacts that were evaluated during the time-line of the contacts was reported according to the Desmond user manual. Furthermore, we improved the captions of the Figures to better understand the meanings.

Unfortunately, the life-time of the interactions is not directly evaluated in terms of ns by the tools available by Desmond. However, by observing the two graphs, it is possible to establish the main type of interactions and the interaction fraction of each type of contact in terms of the percentage with respect to the entire simulation (as mentioned above and inserted in the caption), extrapolating an approximate life-time of the selected interaction.

Secondly, the large blocks of text in several sections, such as Introduction, Discussion and Materials and Methods (4.6) could be divided into smaller paragraphs to improve reading experience.

Authors: In accordance with this comment, we have divided the main sections of the manuscript into smaller paragraphs. Furthermore, to improve the readability of the manuscript, in the Results section, we have introduced subheadings related to each hit compound presented in the main text.

Reviewer 2 Report

Comments and Suggestions for Authors

The manuscript describe the virtual screeneing for new inhibitors for Mac1 domain of SARS-CoV2.

Remarks:

1. Reference 10 is about SARS-CoV, not SARS-CoV2.

2. line 129: idid not understand what means "glycine-rich residues"

3. Several selected compounds have adenyl fragment, but according to fig. 2 and 3, this fragment interact with protein in diffrernt poses than in crystal structure. Why do you think, that selected poses are correct.

4. Method of molecular dynamics is well-known method. Thus I think that description of in in METHODS may be shorter.

5. Plot of RSMD for several ligands have shown that ligands moved diring simulation of MD. And binding energy must be calculate on the part of trajectory on the plateau of RMSD.

6. line 448: "The new theories suggest that the old lock and key theory..." this "new theory" more than 60 years.

Author Response

Reviewer 2

The manuscript describe the virtual screeneing for new inhibitors for Mac1 domain of SARS-CoV2.

Authors: We sincerely thank the reviewer for the positive evaluation of our manuscript. The remarks have been addressed in the revised version of the manuscript.

Remarks:

  1. Reference 10 is about SARS-CoV, not SARS-CoV2.

Authors: Dear reviewer, we have replaced reference 10 as suggested. Thank you for your detailed revision: new citation number 10 is Rezaei S, Sefidbakht Y. Structure of SARS-CoV-2 proteins. COVID-19: Science to Social Impact. 2021:91-120.

  1. line 129: idid not understand what means "glycine-rich residues"

Authors: Thank you for highlighting this aspect. To clarify the meaning of glycine-rich region, we have removed the term residues replacing by region and have added the following sentence to the section to provide a better understanding of this term: “The glycine-rich region is a particular region of the protein with higher flexibility because of the presence of several glycine residues.”

  1. Several selected compounds have adenyl fragment, but according to fig. 2 and 3, this fragment interact with protein in diffrernt poses than in crystal structure. Why do you think, that selected poses are correct.

Authors: We believe that this difference for compound NSC20272 occurred depending on the output exported from the MVD software. To ensure that we will see the same interaction or not as you highlighted, we used new output files for NSC20272 and NSC358078 again and visualized them using Discovery studio software. We observed a new 2D interaction that perfectly shows the same residues that interact with the adenyl group of the ADP-ribose molecule as well as the adenyl groups of these two compounds. Therefore, we updated Figure 3 according to the 2D interactions and added the following sentence to the Results section to further highlight the importance of these amino acids in the interaction with the adenyl fragments of the compounds: “The adenyl fragments in compounds NSC20272 and NSC358078 were able to interact with Asp22, Ile23, and Phe156 through H-bonds, which also interact with the adenyl group of the ADP-ribose molecule.”

  1. Method of molecular dynamics is well-known method. Thus I think that description of in in METHODS may be shorter.

Authors: Thank you for your suggestions. In the experimental section, we have shortened the description of molecular dynamics accordingly.

  1. Plot of RSMD for several ligands have shown that ligands moved diring simulation of MD. And binding energy must be calculate on the part of trajectory on the plateau of RMSD.

Authors: As argued by the reviewer, the three compounds reported in the Supplementary Materials file showed limited binding stability because of the conformational changes that occurred in the ligands. This is reflected in the high RMSD and the impossibility of reaching stability inside the selected binding site. This is particularly indicative of a ligand that is not efficient in binding the protein and has limited affinity. Accordingly, the ligand energy was determined for the entire trajectory using the Desmond script. In this way, all conformations contribute to the calculation of affinity (positive and negative conformations in terms of energy contribution to the total ligand energy), which reflects the difficulty of the ligand to reach a stable bioactive conformation within the selected binding site, suggesting a reduced affinity. Furthermore, we would like to highlight that before starting MD, the system is already minimized and equilibrated by the equilibration of the system approach provided in the Desmond protocol. This step consists of a series of restrained minimization and MD simulations applied to slowly relax the system. Accordingly, when MD starts, all frames are suitable for calculating the binding free energy. Therefore, we used the entire trajectory to perform this calculation as described by us and other scientists (10.1007/s11030-023-10650-6; 10.1080/07391102.2020.1824814; 10.3390/ijms24087578, only for citing some recent examples).

  1. line 448: "The new theories suggest that the old lock and key theory..." this "new theory" more than 60 years.

Authors: Thank you for your detailed perspective. We have replaced the word “new” with “previous theories” to avoid any mistake.